# PRIVACY PRESERVING GENERATIVE FEATURE TRANSFORMATION

## ABSTRACT

Data-Centric AI (DCAI) aims to use AI to get better data for better AI. Feature transformation, as one of the essential tasks of DCAI, can augment the data representation and has garnered significant attention. Existing methods have demonstrated state-of-the-art performance on advancing predictive tasks. However, these methods can lead to serious privacy leakage. For example, sensitive features in original data can be inferred by models trained on transformed data, exposing vulnerabilities in the privacy-preserving capabilities of these methods. To address this issue, we introduce a privacy-preserving feature transformation framework that transforms data representation while preserving privacy from a generative modeling perspective. Specifically, our framework includes two phases: 1) privacy-aware knowledge acquisition and 2) privacy-preserving feature space generation. In the knowledge acquisition phase, we develop an information bottlenecks guided reinforcement learning system to explore and collect privacy-aware feature sets as a knowledge base in token sequence form. In the feature space generation phase, we develop a generative model to encode the knowledge base into a privacy-aware latent space, where the best latent representation is identified and decoded into the optimal privacy-preserving feature space. We solve the optimization via projected gradient ascent that maximizes predictive performance and minimizes privacy exposure. Finally, we present extensive experiments on eight real-world datasets to evaluate how our method can navigate both performance and privacy. The code is available at https://anonymous.4open.science/r/anonymous-2B53/.

## 1 INTRODUCTION

Data-Centric AI (DCAI) aims to use AI to get better data, instead of model architectures, for better AI and has garnered significant attention (Zha et al., 2023). One essential task within DCAI is feature transformation, which involves altering or creating new features from existing data to better represent underlying patterns (Wang et al., 2022; 2024a). However, existing methods of feature transformation can expose sensitive information and lead to serious privacy leakage. For example, our preliminary analysis in Appendix E.1 uncovers that: sensitive attributes (e.g., demographic features), although deleted intentionally by data owners, can still be inferred by other features, or from models trained on transformed data. This example exposes vulnerabilities in privacy-preserving capabilities in feature transformation. As data regulations become increasingly stringent, such as the General Data Protection Regulation in Europe (Voigt & Von dem Bussche, 2017), there is a pressing need for integrating privacy-preserving with feature transformation to safeguard sensitive information while still augmenting data's AI power. In this paper, we research the AI task of privacy-preserving feature transformation, which refers to techniques that transform feature space from original data to advance data's AI readiness while preserving privacy.

Prior literature on feature transformation is two-fold: (1) Search in discrete space: such methods regard feature transformation as a discrete space search problem and solutions are based on a smart search of optimal combinations of feature crosses, for instance, exhaustive expansion then reduction (Katz et al., 2016), iterative-greedy (e.g., Autocross) (Dor & Reich, 2012), evolutionary algorithms (e.g., Genetic Algorithm) (Zhu et al., 2022a). (2) Optimization in continuous space: such methods represent a feature set as an embedding vector, then identify the optimal embedding point in such embedding space, and finally reconstruct the optimal feature set as the target of feature transformation (Ying et al., 2023; Wang et al., 2024a). However, most of these methods focus on augmenting data predictive power and lack privacy considerations. This limits the applicability of feature transformation in privacy-sensitive areas, such as healthcare and education. Privacy-preserving feature transformation is proposed to fill this gap.

There are two major challenges in achieving our goal: 1) acquiring knowledge of and 2) learning knowledge of privacy-preserving feature sets. Firstly, it is difficult to describe the generalizable patterns of a privacy-preserving feature set. There is limited data that encode such knowledge. Knowledge acquisition is to automatically build a knowledge base (i.e., training data) of diverse feature sets with strong/weak privacy-preserving and predictive capabilities in a machine-learnable form. Secondly, after building a knowledge base, we need a new machine learning paradigm to optimize both predictive power and privacy preservation in feature transformation.

**Our Perspective: navigating data privacy and augmentation in feature transformation via progressively tightening constraint-based optimization.** LLMs model world knowledge as sequential tokens and convert question answering into an optimizable generative task in a continuous embedding space. This insight inspires us to treat a transformed feature set as a sequence of feature-feature cross tokens (e.g., $f_1 + f_2, f_3/f_4, ...$), thereafter feature transformation can be seen as a generative task that encodes historical feature transformation knowledge into a latent space, identifies the representation of the best transformed feature set, and reconstructs the optimal transformed features. This is a flexible and optimizable paradigm consisting of model architecture, objective function, and gradient-based optimization. With such a paradigm, we can measure and integrate immeasurable privacy awareness and feature transformation as one through information bottleneck theory and progressively tightening constraint-based optimization.

**Summary of Proposed Method.** Inspired by these insights, we develop a generic and principled privacy-preserving generative feature transformation framework by blending the power of generative AI, privacy information bottleneck, and progressively tightening constraint-based optimization. This framework includes two phases: (1) privacy-aware knowledge acquisition and (2) privacy-preserving feature space generation. To achieve knowledge acquisition, we develop information bottleneck (Tishby et al., 2000; Tishby & Zaslavsky, 2015) guided multi-agent reinforcement learning to explore and collect privacy-aware transformed feature sets. The reinforcement agents maximize the mutual information between transformed features and downstream tasks and minimize the mutual information between transformed features and sensitive features. The explored feature sets are seen as a knowledge base of patterns with various privacy and accuracy scores. To achieve privacy-aware generative transformation, we regard a feature set as a sequence of tokens and map it into a latent representation in a latent space via a sequential encoder. We devise two evaluators to respectively estimate the downstream task performance and privacy exposure risk of the feature set, in order to form optimization objectives and constraints. We identify the best representation of a feature set via progressively tightening constraint-based gradient ascent and leverage a sequential decoder to decode the optimal representation into the optimal feature set. Extensive experiments quantify the effectiveness of our method and demonstrate the privacy awareness of the generated new features in a fine-grained. For example, our proposed method is 7.48% higher than the strongest baseline in terms of comprehensive metric on the Housing Boston dataset.

**Our contributions** are: 1) *AI Task*: We formulate a generic and important task: privacy-preserving feature transformation that navigate privacy and performance in feature transformation in the contexts of data augmentation. 2) *Framework*: We develop an acquisition-generation framework for learning to generate privacy-preserving feature spaces from a generative model perspective. 3) *Computing*: We design interesting techniques to address computing issues. In the generation phase, we integrate generative learning with progressively tightening constraint optimization to trade off privacy and performance. 4) *Data*: In the acquisition phase, we develop information bottleneck guided reinforcement learning as automated knowledge acquisition to measure unmeasurable privacy and acquire privacy-aware feature transformations as training data.

## 2 PROBLEM STATEMENT

Our research problem is to transform the original feature space into a new feature space that further improve the performance of downstream tasks while avoiding the exposure of sensitive features in a traceable and interpretable way. Formally, given the dataset $\mathcal{D} = \{F, s, y\}$, where $F$ is the original feature set (i.e., feature space) consisting of a set of features $f$; $s$ is a sensitive feature involving privacy, which used in the transformation, but not directly utilized for the prediction; and $y$ is the target label. We use $A_{pe}$ to refer to the downstream task model, $A_{pr}$ to refer to the model that predicts sensitive features, and $O$ to refer to the entire set of operators (e.g., "square," "exp," "plus," "multiply," etc.). Our task is to construct the new feature space $\hat{F}$ and identify the ideal one $F^*$ in reconstruction. The optimization objective can be formulated as follows:

$$F^* = \begin{cases} \arg\max_{\hat{F}} \left( \mathcal{L}(A_{pe}(\hat{F}); y) \right) \\ \arg\min_{\hat{F}} \left( \mathcal{L}(A_{pr}(\hat{F}); s) \right). \end{cases} \quad (1)$$

# 3 METHODOLOGY

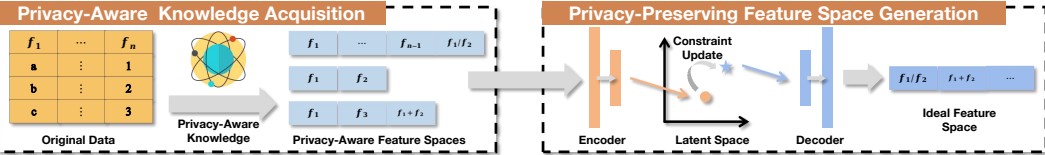

Figure 1: Framework Overview

## 3.1 FRAMEWORK OVERVIEW

In this paper, we propose the **P**rivacy-preserving generative **F**eature **T**ransformation (PFT). Figure 1 shows the framework of PFT including two main steps: 1) privacy-aware knowledge acquisition; and 2) privacy-preserving feature space generation.

In the knowledge acquisition phase, we use multi-agent reinforcement learning to implement the selection of candidate features and candidate operations for feature crossing. Information bottleneck is used to guide the decision-making process of agents. We minimize the mutual information between the new feature space and downstream tasks while minimizing the mutual information between the new feature space and sensitive features. Collected privacy-aware feature sets account for both privacy and performance, which then are serialized as a knowledge base.

In the feature space generation phase, we map the knowledge base into a privacy-aware latent space by a sequence encoder. Two evaluators are used to estimate the performance on downstream tasks and the risk of exposing sensitive information of a transformed feature set using the latent representation. We use estimates of downstream task performance to provide gradient guidance, and estimates of risk of privacy exposure to provide gradually tightening constraints. Finally, a sequence decoder is used to decode the updated latent representation.

## 3.2 PRIVACY-AWARE KNOWLEDGE ACQUISITION

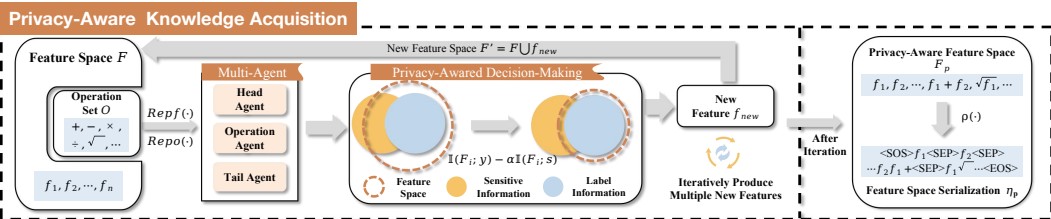

Figure 2: Privacy-Aware Knowledge Acquisition (Phase 1)

### 3.2.1 MULTI-AGENT REINFORCEMENT LEARNING

Multiple interdependent Markov Decision Processes (MDPs) can effectively describe the construction of new features (Wang et al., 2022; Xiao et al., 2023). We aim to construct feature sets with privacy-aware knowledge in this way to provide high-quality data for subsequent generative models. We decompose this process into three MDPs using a cascading structure of three reinforcement learning agents., including two MDPs for picking features, and one MDP for picking operators.

**State Representation** $Repf(\cdot)$ **&** $Repo(\cdot)$: We first represent the features and operators to facilitate model processing. For features, we employ a descriptive statistical technique $Repf(\cdot)$ to obtain this state representation (Heaton, 2016). In detail, we first compute the feature set column-wise descriptive statistics (i.e., count, standard deviation, minimum, maximum, first, second, and third quantile). Then, we calculate the same descriptive statistics on the output of the previous statistics. After that, we can obtain the descriptive matrix and flatten it as the state representation. For the representation of the operator, we pre-determine the types of operations available and then use a one-hot encoding $Repo(\cdot)$ to get a representation of the operator.

**Reinforcement Learning Agents**: We use the classic DQN structure to implement agents (Mnih et al., 2015). We adopt the $i^{th}$ iteration as an example to describe the cooperation between agents. First, the *head feature agent* selects feature $f_h$ as the header feature based on the $(i-1)^{th}$ iteration's feature space state representation $Repf(F_{i-1})$, then the *operator agent* selects operator $o_i$ based on feature space and header feature $Repf(F_{i-1})||Repf(f_h)$, where $||$ indicates concatenation. Finally, the *tail feature agent* selects tail feature $f_t$ as the tail feature based on feature space, header feature and operator $Repf(F_{i-1})||Repf(f_h)||Repo(o_i)$. New features $f_i$ are obtained by calculating the head and tail features according to the operator. The $(i-1)^{th}$ iteration's feature space $F_{i-1}$ combines with new features $f_i$ to be the new feature space $F_i$.

### 3.2.2 PRIVACY-AWARED DECISION-MAKING

Feedback-based policy learning is used to optimize each agent to find privacy-aware features. Ideally, the privacy-aware features should improve performance on downstream tasks, and avoid exposure to sensitive features. Consistent with previous literature (Wang et al., 2022; 2024a), we consider all features as an entire feature space to avoid the negative impact (shown in the Appendix E.1) of complex interdependencies between features, so that sensitive features can be used to produce valuable new features in the transformation process without further exposure. We design a privacy-awared reward function $\mathcal{R}(\cdot)$ to guide agents' decision-making according to the information bottleneck principle (Tishby et al., 2000; Tishby & Zaslavsky, 2015). We design the reward function from two aspects: (1) maximize the mutual information between the new feature space and the downstream task label, and (2) minimize the mutual information between the new feature space and the sensitive feature.

$$\mathcal{R}(F_i, y, s) = \mathbb{I}(F_i; y) - \alpha\mathbb{I}(F_i; s), \tag{2}$$

where $\mathbb{I}(\cdot; \cdot)$ denotes mutual information, $y$ denotes the groundtruth of the downstream task, $s$ denotes the sensitive feature.

**Maximize Mutual Information Lower Bound**: By maximizing mutual information, we encourage the construction of new feature spaces that can enhance downstream tasks.

$$\mathbb{I}(F_i; y) \stackrel{(a)}{=} H(F_i) - H(F_i|y) \stackrel{(b)}{\geq} -H(F_i|y) \stackrel{(c)}{\geq} \sum p(F_i) \log\left(p(F_i|y)\right)$$
$$\stackrel{(d)}{\geq} \log\left(p(F_i|y)\right) \stackrel{(e)}{=} \log\left(\phi(\mathcal{D}(F_i))\right) \stackrel{(f)}{\geq} \log\left(\phi(\mathcal{D}(F_i))\right) - \log\left(\phi(\mathcal{D}(F_{i-1}))\right), \tag{3}$$

where $H(\cdot)$ refers to the information entropy, $\mathcal{D}(\cdot)$ denote the model of downstream task, $\phi(\cdot)$ is the sigmoid activation. In the above derivation, (a) is the definition of mutual information; (b) is the non-negative property of $H(F_i)$; (c) is the definition of information entropy; (d) is that $\sum p(F_i) \leq 1$; (e) $\phi(\mathcal{D}(F_i))$ is the variational approximation of $p(F_i|y)$; (f) is because $\mathcal{D}(F_{i-1})$ is a non-negative constant, and through experiments, we found that using the increments in downstream task performance, rather than the performance itself, provides clearer guidance to the model. Finally, we maximize the incremental performance of the feature space generated by two iterations on the downstream task to maximize the mutual information between the constructed feature space and the downstream task.

**Minimize Mutual Information Upper Bound**: Considering only the performance, there is a risk of exposing sensitive information. Therefore, we minimize the mutual information between new feature space and sensitive features to provide privacy-awared decision-making guidance for agents. However, estimating the upper bound of mutual information is an intractable problem. Although some studies leverage variational techniques to estimate the upper bound, they heavily rely on the prior assumption (Alemi et al., 2016; Cheng et al., 2020). Therefore, refer to prior works (Ma et al., 2020a; Yang et al., 2024), we introduce the Hilbert-Schmidt Independence Criterion (HSIC) (Gretton et al., 2005b) as the approximation of the minimization of $\mathbb{I}(F_i; s)$.

HSIC serves as a statistical measure of dependency, which is formulated as the Hilbert-Schmidt norm, assessing the cross-covariance operator between distributions within the Reproducing Kernel Hilbert Space (RKHS). Given $F_i$ and $s$, $HSIC(F_i, s)$ is defined as follows:

$$HSIC(F_i; s) = \|C_{F_i s}\|_{hs}^2$$
$$= \mathbb{E}_{F_i, F_i', s, s'}[K_{F_i}(F_i, F_i')K_s(s, s')] \tag{4}$$
$$+ \mathbb{E}_{F_i, F_i'}[K_{F_i}(F_i, F_i')] - 2\mathbb{E}_{F_i, s}[K_{F_i}(F_i, F_i')][K_s'(s, s')]$$

where $C_{F_i s}$ is the cross-covariance operator between the Reproducing Kernel Hilbert Spaces (RKHSs) of $F_i$ and $s$, $\|\cdot\|_{hs}^2$ refers to the Hilbert-Schmidt norm, $K_{F_i}$ and $K_s$ are two kernel func-

tions for variables $F_i$ and $s$, $F_i'$ and $s'$ are two independent copies of $F_i$ and $s$. Given the sampled instances $(F_{i_j}, s_j)_{j=1}^n$ from the batch training data, we estimated HSIC as:

$$H\hat{S}IC(F_i; s) = Tr(K_{F_i}HK_sH)(n-1)^{-1}, \tag{5}$$

where $K_{F_i}$ and $K_s$ are used kernel matrices (Gretton et al., 2005a), with elements $K_{F_{i_{jj'}}} = K_{F_i}(F_{i_j}, F_{i_{j'}})$ and $K_{s_{jj'}} = K_s(s_j, s_{j'})$, $H = \mathbf{I} - \frac{1}{n}\mathbf{1}\mathbf{1}^T$ is the centering matrix, and $Tr(\cdot)$ denotes the trace of matrix. In practice, we adopt the widely used radial basis function (RBF) (Vert et al., 2004) as the kernel function:

$$K_{F_i}(F_{i_j}, F_{i_{j'}}) = \exp-\frac{\|F_{i_j} - F_{i_{j'}}\|^2}{2\sigma^2} \tag{6}$$

where $\sigma$ is the parameter that controls the sharpness of RBF. In order not to rely on prior assumptions and to calculate more efficiently, we minimize $H\hat{S}IC(F_i; s)$ instead of minimizing $\mathbb{I}(F_i; s)$.

Finally, we use the reward function Equation (7) to guide agents to construct new feature spaces that benefit downstream tasks while avoiding sensitive feature exposure:

$$\mathcal{R}(F_i, y, s) = \mathbb{I}(F_i; y) - \alpha H\hat{S}IC(F_i; s). \tag{7}$$

**Feature Space Serialization:** After collecting privacy-aware feature spaces, we represent these privacy-aware feature spaces as a sequence $\eta_p$ by the convert function $\rho(\cdot)$. In detail, we encode all the features and all the operators in a unified token space. For the new feature generated by the original feature, we use Reverse Polish Notation (Łukasiewicz, 1957) to represent its generation path. Because of the uniqueness and extensibility of the Reverse Polish Notation, we can encode and optimize more accurately and conveniently. Besides, three special tokens are introduced: $\langle SEP \rangle$, $\langle SOS \rangle$, and $\langle EOS \rangle$, respectively, to mark the split between features, the beginning and end of a feature space. For each feature space from the knowledge base, we perform data augmentation by randomly shuffling the order of features. The detailed pseudo code of this conversion process and an example of conversion are provided in Appendix A.

### 3.3 PRIVACY-PRESERVING FEATURE SPACE GENERATION

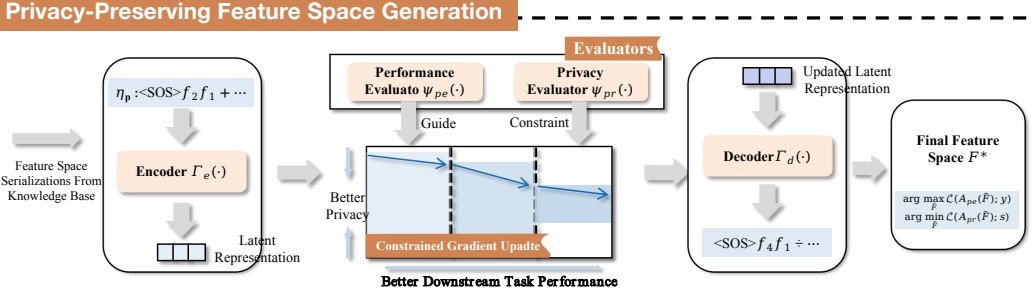

Figure 3: Privacy-Preserving Feature Space Generation (Phase 2)

Supported by rich privacy-aware knowledge, we use generative models to achieve more stable and robust feature space generation (Wang et al., 2024a). We use an autoencoder structure to map the feature space in the knowledge base to the latent space and find better points in the latent space guided by the performance of downstream tasks with progressively tightening privacy constraints.

#### 3.3.1 SEQUENCE AUTOENCODER STRUCTURE

We use serialized feature spaces $\eta_p = \rho(F_p)$ as privacy-aware knowledge for training encoder $\Gamma_e(\cdot)$ and decoder $\Gamma_d(\cdot)$ to obtain a desired latent space. We adopt a single layer long short-term memory (LSTM) (Hochreiter & Schmidhuber, 1997) as encoder $\Gamma_e(\cdot)$ and we acquire the continuous latent representation $E_p$ of the feature space $F_p$, denoted by $E_p = \Gamma_e(\rho(F_p))$. We adopt a single layer LSTM as decoder $\Gamma_d(\cdot)$. The decoder decodes latent representation into Reverse Polish Notation $\hat{\eta}_p$ in a sequence-to-sequence way (Sutskever et al., 2014). Given the latent representation $E_p$, to make the generated sequence similar to the real one, we minimize the negative log-likelihood of the distribution, defined as: $\mathcal{L}_{rec} = -\log P_{\Gamma_d}(\hat{\eta}_p; E_p)$.

### 3.3.2 PERFORMANCE AND PRIVACY EVALUATORS

To generate the ideal feature space, we first organize the latent space for targeted optimization. Two evaluators are employed to clarify the relationship between latent representations, downstream task performance, and sensitive features. In particular, the performance evaluator $\Psi_{pe}(\cdot)$ models the relationship between latent representations and downstream task performance, which is then used to provide the optimization objective to update latent representations for better downstream performance. The privacy evaluator $\Psi_{pr}(\cdot)$ models the relationship between latent representation and privacy exposure risk, which is then used to provide constraints to keep sensitive features secure.

**Performance Evaluator:** We expect the latent representation to indicate the accuracy of the corresponding feature space on the downstream task so that we can obtain a higher performance feature space by purposefully adjusting the latent representation. We use a performance evaluator to establish this relationship, denoted as $\hat{v} = \Psi_{pe}(E_p; y)$. We use a simple linear layer to implement $\Psi_{pe}(\cdot)$. We train the parameters $\Psi_{pe}$ of the estimator by minimizing the Mean Squared Error (MSE) between the estimate and the true value $\min_{\Psi_{pe}} \mathcal{L}_{pe} = MSE(v|\hat{v})$.

**Privacy Evaluator:** Similarly, we can use the privacy evaluator $\Psi_{pr}(\cdot)$ to assess the extent to which latent representations reveal sensitive features. According to Section 3.2.2, we leverage HSIC to describe the relationship between feature space and privacy. The privacy estimators estimate the HSIC of latent representations, given as $H\tilde{S}IC = \Psi_{pr}(E_p; s)$. We also use a simple linear layer to implement $\Psi_{pr}(\cdot)$. We train the parameters $\Psi_{pr}$ of estimator by minimizing the MSE between the estimate and the true value $\min_{\Psi_{pr}} \mathcal{L}_{pr} = MSE(HSIC(F_p; s); H\tilde{S}IC)$.

We use a multi-tasking architecture to train an autoencoder structure with two evaluators:

$$\mathcal{L} = \mathcal{L}_{rec} + \mathcal{L}_{pe} + \mathcal{L}_{pr}, \tag{8}$$

### 3.3.3 CONSTRAINED GRADIENT UPDATE

After the encoder and two evaluators are jointly trained, each latent representation (1) can reconstruct the feature space; (2) can reflect the performance of the corresponding feature space in the downstream task; (3) can reflect the degree of exposure of sensitive features.

On this basis, we optimize the latent representation to further improve the accuracy of downstream tasks while ensuring privacy. To alleviate the problem of difficulty in training and balancing caused by dual objectives, we distinguish the roles of the two objectives. Performance is used as the optimization goal, and privacy is used as a gradually tightened constraint. The initial constraint allows the model to better inherit privacy-aware knowledge, and the gradually tightened constraint allows the model to focus on performance while also strengthening privacy. Specifically, for the latent representation $E_p$, under the constraints of the privacy evaluator $\Psi_{pr}(E_p; s)$, we search toward the gradient direction induced by the performance evaluator $\Psi_{pe}(E_p; y)$:

$$\hat{E}_p = E_p + \eta \frac{\partial \Psi_{pr}}{\partial E_p}$$
$$\text{s.t.} \quad \Psi_{pr}(\hat{E}_p; s) \leq \Psi_{pr}(\hat{E}_p^{min}; s), \tag{9}$$

we perform this search $T$ times to get $\{\hat{E}_p^1, \ldots, \hat{E}_p^T\}$ and $\hat{E}_p^{min}$ is the result with the best privacy evaluated in the previous search. With continuous iterations, $\hat{E}_p^{min}$ will gradually become smaller, and the model needs to meet increasingly tighter privacy constraints. In the implementation, we use projected gradient ascent (Madry, 2017) to implement this constraint. We select multiple $E_p$ as seeds for the search and use beam search strategy (Sutskever, 2014) to determine the best result $\hat{E}_p^*$. The best updated latent representation is decoded by the decoder to the final feature space $F^*$. The implementation details can be found in Appendix C.

## 4 EXPERIMENT

### 4.1 EXPERIMENTAL SETUP

#### 4.1.1 DATA DESCRIPTION

We select 4 user-related real datasets (Housing Boston, German Credit, Uci Credit Card, Amazon Employee), which contain sensitive features that can be pointed out. Besides, we select 4 additional

real datasets (Lymphography, Openml 618, Activity, AP Omentum Ovary) and randomly assign a sensitive feature. These datasets cover different domains and scales, covering classification and regression problems. Statistics and detailed descriptions are provided in Appendix B.

### 4.1.2 Evaluation Metrics

For the performance of downstream tasks (DT), we evaluate classification with the F1 score (Powers, 2011) and regression with the 1-Relative Absolute Error (1-RAE) (Wang et al., 2022), aiming for higher values. For privacy, we assess sensitive feature (SF) prediction in a manner consistent with the downstream tasks, using 1-F1 for classification and RAE for regression, with higher values indicating better privacy protection. Since both performance and privacy are equally important in our task, we compute their arithmetic mean (Avg) for a comprehensive comparison.

### 4.1.3 Baseline Methods

We select three types of methods to compare with PFT , including the feature transformation methods: (1) **ORG** denotes use original dataset to predict. (2) **RDG** generates feature-operation-feature transformation records at random ; (3) **ERG** first applies operations to each feature to expand the feature space, then selects the crucial features. (4) **AFAT** (Horn et al., 2020) is an enhanced version of ERG that uses multi-step feature selection to select informative ones. (5) **NFS** (Chen et al., 2019) models the transformation sequence of each feature and uses RL to optimize the entire process. (6) **TTG** (Khurana et al., 2018) formulates the transformation process as a graph, then implements an RL-based search method to search. (7) **GRFG** (Wang et al., 2022) uses three collaborated reinforced agents to conduct feature generation. (8) **MOAT** (Wang et al., 2024a) develops an embedding-optimization-reconstruction framework to produce high-quality feature space. Privacy-preserving methods that are available for our problem, include: **Data Perturbation (DP)** (Dwork et al., 2014) add noise to the sensitive features to perturb them, and then participate in downstream tasks and prediction of sensitive features. The addition of noise complies with the standard of differential privacy (Dwork, 2006). And combination methods, including: (1) **GRFG-DP** uses GRFG for feature transformation and then uses the DP method to process sensitive features. (2) **MOAT-DP** uses MOAT for feature transformation and then uses the DP method to process sensitive features. We mainly use Random Forests (Breiman, 2001) as the model for downstream tasks. Because it is a robust, stable, well-tested method, thus, we can reduce performance variation caused by the model. We provide experimental results on other downstream task models in the Appendix E.3 and d more experimental and hyperparameter settings in Appendix D

### 4.2 Overall Comparison

Table 1: Comparison results on user-related datasets. DT represents the prediction accuracy on downstream tasks, and SF represents the prediction accuracy on sensitive features, and Avg represents the average accuracy.

| Dataset | Housing Boston | | | German Credit | | | Uci Credit Card | | | Amazon Employee | | |
|---|---|---|---|---|---|---|---|---|---|---|---|---|
| Metric | DT ↑ | SF↑ | Avg↑ | DT↑ | SF↑ | Avg↑ | DT↑ | SF↑ | Avg↑ | DT↑ | SF↑ | Avg↑ |
| ORI | 0.4012 | 0.1630 | 0.2821 | 0.7012 | 0.4476 | 0.5744 | 0.7992 | 0.9665 | 0.8829 | 0.9275 | 0.0197 | 0.4736 |
| RDG | 0.4411 | 0.0472 | 0.2442 | 0.7262 | 0.1214 | 0.4238 | 0.9740 | 0.0755 | 0.5248 | 0.9310 | 0.0260 | 0.4785 |
| ERG | 0.4080 | 0.0234 | 0.2157 | 0.7442 | 0.0729 | 0.4086 | 0.8030 | 0.0776 | 0.4403 | 0.9352 | 0.0239 | 0.4796 |
| AFAT | 0.4099 | 0.0359 | 0.2229 | 0.7013 | 0.4392 | 0.5703 | 0.8056 | 0.9565 | 0.8810 | 0.9339 | 0.0381 | 0.4860 |
| NFS | 0.4251 | 0.1433 | 0.2842 | 0.7061 | 0.4780 | 0.5921 | 0.8054 | 0.9531 | 0.8793 | 0.9300 | 0.0459 | 0.4880 |
| TTG | 0.4140 | 0.1712 | 0.2926 | 0.7250 | 0.4499 | 0.5875 | 0.7989 | 0.9609 | 0.8799 | 0.9316 | 0.0366 | 0.4841 |
| GRFG | 0.4212 | 0.1109 | 0.2661 | 0.7187 | 0.4555 | 0.5871 | 0.8050 | 0.9611 | 0.8831 | 0.9309 | 0.0431 | 0.4870 |
| MOAT | 0.4648 | 0.0391 | 0.2520 | 0.7459 | 0.4432 | 0.5946 | 0.8087 | 0.9594 | 0.8840 | 0.9344 | 0.0451 | 0.4898 |
| DP | 0.4079 | 0.1803 | 0.2941 | 0.7080 | 0.4587 | 0.5834 | 0.7936 | 0.9682 | 0.8809 | 0.9249 | 0.0261 | 0.4755 |
| GRFG-DP | 0.4012 | 0.1322 | 0.2667 | 0.7005 | 0.4664 | 0.5835 | 0.7984 | 0.9670 | 0.8827 | 0.9323 | 0.0196 | 0.4760 |
| MOAT-DP | 0.4601 | 0.0691 | 0.2646 | 0.6905 | 0.4582 | 0.5743 | 0.8042 | 0.9637 | 0.8840 | 0.9348 | 0.0516 | 0.4932 |
| Ours | 0.4574 | 0.1747 | **0.3161** | 0.7579 | 0.4698 | **0.6139** | 0.8083 | 0.9745 | **0.8914** | 0.9310 | 0.0747 | **0.5029** |

As shown in Table 1 and 2, we compare our model with other baselines on multiple datasets. We have the following observations:

(1) On all datasets, the average performance of PFT is the best. For example, PFT is 7.48% and 4.70% higher than the strongest baseline in terms of the average metric on datasets Housing Boston and Lymphography, respectively. This demonstrates that our proposed method preserves privacy while enhancing the performance of downstream tasks, which meets the requirement of integrating privacy-preserving with feature transformation.

Table 2: Comparison results on non-user-related datasets.

| Dataset | Lymphography | | | Openml 618 | | | AP Omentum Ovary | | | Activity | | |
|---------|--------|--------|--------|--------|--------|--------|--------|--------|--------|--------|--------|--------|
| Metric | DT ↑ | SF ↑ | Avg ↑ | DT ↑ | SF ↑ | Avg ↑ | DT ↑ | SF ↑ | Avg ↑ | DT ↑ | SF ↑ | Avg ↑ |
| ORI | 0.7175 | 0.4445 | 0.5810 | 0.4120 | 0.0423 | 0.2272 | 0.6124 | 0.5061 | 0.5593 | 0.9503 | 0.0398 | 0.4951 |
| RDG | 0.6850 | 0.3110 | 0.4980 | 0.4700 | 0.0443 | 0.2572 | 0.6512 | 0.4492 | 0.5502 | 0.9555 | 0.0335 | 0.4945 |
| ERG | 0.6850 | 0.1910 | 0.4380 | 0.4621 | 0.0350 | 0.2486 | 0.6621 | 0.0456 | 0.3539 | 0.9543 | 0.0366 | 0.4955 |
| AFAT | 0.6527 | 0.0356 | 0.3442 | 0.4741 | 0.0365 | 0.2553 | 0.6124 | 0.5743 | 0.5934 | 0.9527 | 0.0391 | 0.4959 |
| NFS | 0.7180 | 0.5039 | 0.6110 | 0.4754 | 0.0420 | 0.2587 | 0.6294 | 0.6121 | 0.6208 | 0.9506 | 0.0657 | 0.5081 |
| TTG | 0.7180 | 0.5501 | 0.6341 | 0.4277 | 0.2060 | 0.3169 | 0.6345 | 0.6215 | 0.6280 | 0.9549 | 0.4361 | 0.6955 |
| GRFG | 0.8133 | 0.6323 | 0.7228 | 0.4688 | 0.2312 | 0.3500 | 0.6443 | 0.6236 | 0.6340 | 0.9516 | 0.4559 | 0.7038 |
| MOAT | 0.8185 | 0.5100 | 0.6642 | 0.4957 | 0.0419 | 0.2688 | 0.6713 | 0.6120 | 0.6416 | 0.9541 | 0.4515 | 0.7028 |
| DP | 0.7175 | 0.6289 | 0.6732 | 0.2206 | 0.0363 | 0.1285 | 0.6124 | 0.5439 | 0.5782 | 0.9488 | 0.4545 | 0.7017 |
| GRFG-DP | 0.8138 | 0.2561 | 0.5349 | 0.4388 | 0.2329 | 0.3359 | 0.6401 | 0.6146 | 0.6274 | 0.9518 | 0.4545 | 0.7032 |
| MOAT-DP | 0.8180 | 0.4990 | 0.6585 | 0.4157 | 0.0369 | 0.2263 | 0.6737 | 0.6148 | 0.6443 | 0.9522 | 0.4546 | 0.7034 |
| Ours | 0.7895 | 0.7595 | **0.7568** | 0.5182 | 0.2120 | **0.3651** | 0.6713 | 0.6286 | **0.6500** | 0.9524 | 0.4597 | **0.7060** |

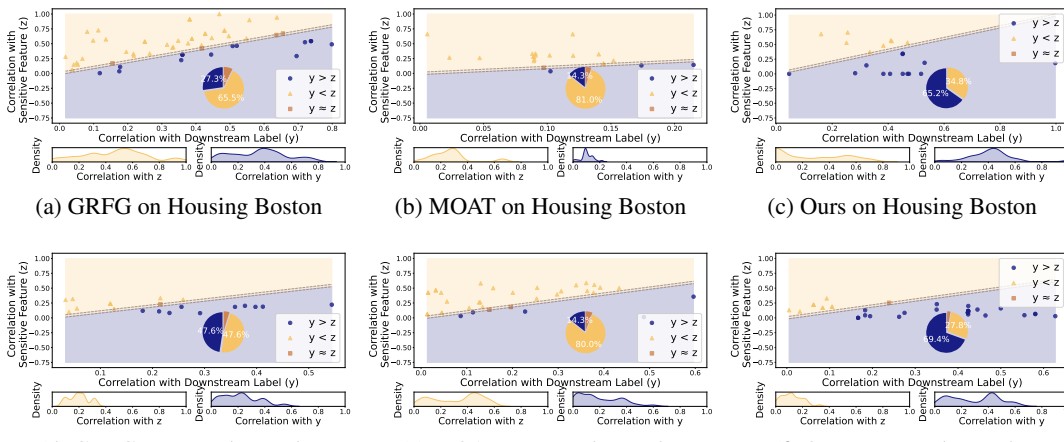

(a) GRFG on Housing Boston     (b) MOAT on Housing Boston     (c) Ours on Housing Boston

(d) GRFG on Lymphography     (e) MOAT on Lymphography     (f) Ours on Lymphography

Figure 4: Correlation of New Features with Label and Sensitive Feature. X-axis is the correlation between the generated feature and the label y. Y-axis is the correlation between the generated feature and the sensitive feature z. The lower area is the correlation between the generated feature and y is higher than the correlation with z. The upper area is the opposite. The pie chart shows the proportion of points in different areas.

(2) We observed that a decline in downstream task performance is not a necessary condition for improving privacy. For instance, on Housing Boston and Openml 618, PFT slightly outperforms the best baseline by about 2% and 4%. We speculate that this may be because the baseline not yet discovering the optimal feature space. While PFT may sacrifice some valuable features that could expose privacy, it is still able to improve performance by leveraging other features as substitutes.

(3) Simply applying some classic privacy protection methods may not be effective. For instance, directly adopting DP does not yield better privacy results on the German Credit dataset. Moreover, simply combining feature transformation methods with privacy techniques also not be effective. For instance, on the German Credit dataset, our method improves privacy protection by 2.5% and overall performance by 6.9% compared to the combination of MOAT and DP. This highlights that privacy protection in feature transformation is a complex issue worth deeper exploration. A straightforward layering of different approaches may not be sufficient to fully address the requirements.

## 4.3 INVESTIGATION OF PFT

### 4.3.1 NEW FEATURE SPACE ANALYSIS

The ideal feature transformation can achieve good performance on downstream tasks while avoiding the exposure of sensitive features. From the perspective of the single feature, we hope that each feature has a strong connection with the downstream task label and little with the sensitive feature. Therefore, we calculate the Pearson correlation coefficient (Cohen et al., 2009) between the feature and the downstream task label y and the sensitive feature z. We take the absolute value of the correlation coefficient and draw a scatter plot and a probability density plot. We divide the scatter

plot into three regions, including y < z, y ≈ z, and y > z. We color the three regions, count the number of features in each region, and draw a pie chart. We hope that the features in the y > z region are the most, which indicates that the model tends to construct features that are beneficial to downstream tasks and do not expose privacy. As shown in Figure 4, compared with the two strongest baselines, PFT is more likely to generate ideal features. For example, on the Lymphography dataset, PFT generates 69.4% of ideal features, which is higher than 47.6% of GRFG and 14.3% of MOAT. Besides, from the probability density plot, we can find that most of the features generated by our method have a correlation close to 0 with z, and also have a strong correlation with y.

### 4.3.2 RELATIONSHIP BETWEEN HSIC AND SENSITIVE INFORMATION

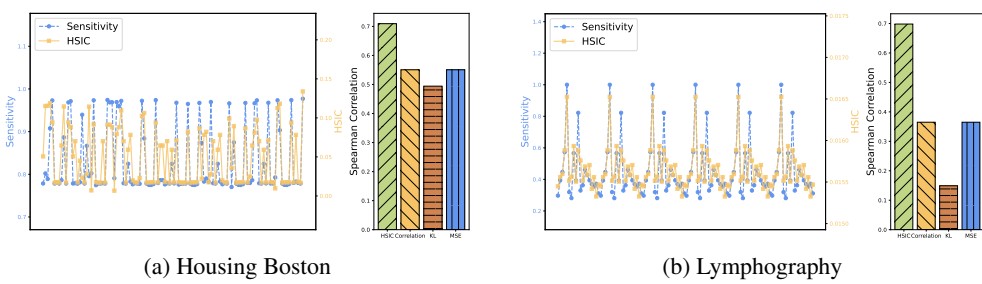

(a) Housing Boston          (b) Lymphography

Figure 5: Relationship Between HSIC and Sensitive Information

We incorporate HSIC into both knowledge acquisition and feature space generation. On the one hand, prior research provided theoretical support for using HSIC as an approximation of the mutual information lower bound (Ma et al., 2020b; Yang et al., 2024). On the other hand, we can avoid the overhead of directly predicting sensitive features. In this section, we analyze the relationship between HSIC and privacy. We generate feature spaces through knowledge acquisition, using task performance as a reward, and examine both the prediction accuracy of sensitive features and the HSIC between feature spaces and sensitive features. As illustrated in the line graph in Figure 5, the trends of HSIC and the prediction accuracy overlap significantly, which shows that HSIC can reflect exposure of sensitive features. The bar chart on the right provides a more intuitive visualization. We calculate the Pearson correlation coefficient, KL divergence (Kullback & Leibler, 1951), and MSE between feature spaces (after pooling) and sensitive features. We compare Spearman's rank correlation coefficient (Spearman, 1961), which measures their monotonic correlation, between these metrics and the prediction accuracy. As shown in the figure, HSIC exhibits the highest Spearman's rank correlation coefficient, indicating that it is consistent with the monotonicity of prediction accuracy. This suggests that HSIC can serve as an effective guide and constraint for privacy considerations.

### 4.4 ABLATION EXPERIMENT AND SENSITIVITY ANALYSIS

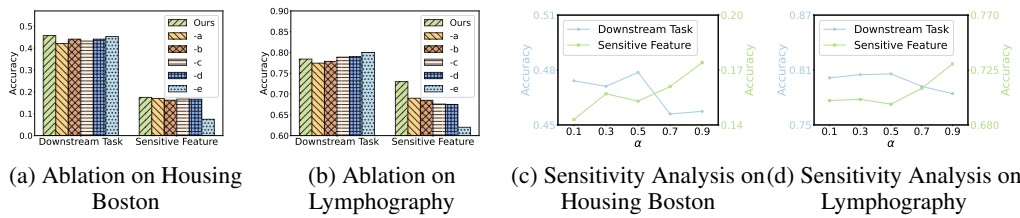

(a) Ablation on Housing Boston    (b) Ablation on Lymphography    (c) Sensitivity Analysis on Housing Boston    (d) Sensitivity Analysis on Lymphography

Figure 6: Ablation Study and Sensitivity Analysis

To explore the effect of each component in PFT , we conducted ablation experiments. PFT-$a$ means that we directly use the knowledge acquisition for evaluation. PFT-$b$ means that we randomly generate feature space as input of the generative model instead of acquiring knowledge. PFT-$c$ means that we only maximize the mutual information between the feature space and the downstream task label in knowledge acquisition. PFT-$d$ means we update the latent representation using the initial privacy constraints without tightening. PFT-$e$ means that we do not distinguish the roles of performance and privacy and optimize both objectives simultaneously. As shown in Figure 6(a) and 6(b), in our method, each module contributes to the final good performance. The two steps are tightly coupled and synergistically improve the two objectives. In particular, PFT-$e$ shows that our distinction between the roles of the objectives is reasonable. It is difficult to achieve such a composite goal

by optimizing both objectives simultaneously. In this case, the generative model even ignores the knowledge provided by the paradigm feature space, further exposing sensitive features.

The hyperparameter $\alpha$ in the model is used to balance the relationship between maximizing the mutual information with the downstream task label and minimizing the mutual information with the sensitive features. We use $\alpha$ to observe the changes in different indicators of the model. In general, as $\alpha$ increases, the knowledge collected is more inclined to protect privacy rather than improve performance. This will lead to a decrease in downstream tasks and better privacy protection effects. Besides, to further analyze PFT, we provide more experimental results (time and space complexity, estimator performance, etc.) in Appendix E

## 5 RELATED WORK

### 5.1 FEATURE TRANSFORMATION

As one essential task within DCAI (Zha et al., 2023; Wang et al., 2024b;a), feature transformation aims to enhance the feature space by generating new features in an explainable and traceable way, thereby improving the performance of machine learning models. Existing methods primarily focus on boosting downstream task performance and can be broadly divided into two categories: (1) Search in discrete spaces (Horn et al., 2020; Wang et al., 2022; Kanter & Veeramachaneni, 2015; Khurana et al., 2016; 2018; Tran et al., 2016; Xiao et al., 2023; Lam et al., 2017; Zhu et al., 2022a; Xiao et al., 2024): These methods treat feature transformation as a discrete space search problem and solutions are based on smart search of optimal combinations of feature crosses. Some works initially add new features to expand the feature space and eventually select only the high-value features to form the final feature set (Katz et al., 2016). Some works adopt an iterative-greedy strategy (Dor & Reich, 2012). Effective features are iteratively generated, and significant ones are preserved until the maximum number of iterations is reached. Some methods combine evolutionary algorithms to explore effective feature spaces (Gong et al., 2024). (2) Optimization in continuous space (Zhu et al., 2022b; Wang et al., 2024a; Ying et al., 2023; 2024): Such methods represent a feature set as an embedding vector in a feature set embedding space, then identify the optimal embedding point in such embedding space, and finally reconstruct the optimal feature set. However, these methods focus on augmenting data predictive power and lack privacy considerations.

### 5.2 INFORMATION BOTTLENECK PRINCIPLE

The Information Bottleneck (IB) method is a principle from information theory used to find an optimal balance between compression and prediction (Tishby et al., 2000; Tishby & Zaslavsky, 2015). This principle has been employed to enhance interpretability and disentangle representations (Bao, 2021; Jeon et al., 2021). However, calculating mutual information between high-dimensional variables is challenging. To address this, researchers have used neural networks to approximate and estimate mutual information (Alemi et al., 2016; Belghazi et al., 2018; Cheng et al., 2020; Oord et al., 2018). However, they relatively rely on the prior assumption and the quality of sampling influences the accuracy of the estimation. Instead of directly optimizing, the Hilbert-Schmidt Independence Criterion (HSIC) has been employed as an alternative to assess variable (Gretton et al., 2005b). Given the challenges in estimating the upper bound of mutual information, we opt for HSIC to approximate and minimize the mutual information between learned representations and sensitive features (Ma et al., 2020a; Wang et al., 2021; Yang et al., 2024; Xie et al., 2024).

## 6 CONCLUSION AND FUTURE WORK

In this paper, we propose a privacy-preserving feature transformation framework called PFT. Specifically, we first use information bottlenecks to guide multi-agents to generate privacy-aware feature sets. We then serialize them as a knowledge base to provide privacy-aware knowledge. Then we map the knowledge in latent space through generative models. We set two evaluators to evaluate the performance and privacy of representations in such latent space and then use them as objective and constraint to find better representations for decoding. Through the proposed framework, we achieve the dual goals of performance and privacy. We use sensitive features to transform features to generate valuable new features without further exposing sensitive information. This has practical significance for the application of data-driven AI systems in sensitive fields. Extensive experiments verify the effectiveness of our model. However, there are still some limitations in our approach, which we will further explore in future work. First, there are multiple ways to calculate HSIC and its relationship with privacy needs further exploration. Additionally, while we hypothesize that performance and privacy can improve simultaneously, more in-depth analysis is required.

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

## A  FEATURE SPACE SERIALIZATION METHODS

In order to use the model to optimize the feature space, we first need to represent it. We treat the feature space as a string. Since the Reverse Polish Notation has the advantages of uniqueness and not relying on brackets for calculations, we first convert the feature space string into Reverse Polish Notation, which makes it easier for the model to recognize and generate unambiguous strings while reducing computational overhead. The conversion of Reverse Polish Notation can be described by the Shunting Yard Algorithm (Dijkstra, 1961), which was designed by Edsger Dijkstra.

We follow the steps below to convert: We first initialize a list of original expressions of feature space and two stacks, $S_1$ and $S_2$, respectively. For each element in the list, we scan it from left to right. When getting a feature name token, we push it to $S_2$. When receiving a left parenthesis, we push it to $S_1$. When obtaining any operations, we pop each element in $S_1$ and push them into $S_2$ until the last component of $S_1$ is the left bracket. Then we push this operation into $S_1$. When getting a right parenthesis, we pop every element from $S_1$ and then push into $S_2$ until we confront a left bracket. Then we remove this left bracket from the top of $S_1$. When the end of the input encounters, we append every token from $S_2$ into the final expression $\eta_p$. If this feature is not the last element in the current feature space, we will append a $\langle SEP \rangle$ token to indicate the end of this feature expression. After we process every element in the features, we add $\langle SOS \rangle$ and $\langle EOS \rangle$ tokens to the beginning and end of the expression of feature space to form the Reverse Polish Notation transformation sequence. Each element in $\eta_p$ is a feature name token, operation token, or three other special tokens. We convert each transformation sequence through this algorithm and construct the training set with them.

We give some examples to illustrate this conversion process. For example, if initial transformed feature spaces as follows:

1 $f_0 * f_4, f_3 + (f_7 * f_1), f_2 \% f_6, f_1 / f_9, (f_5 - f_8) * f_4$

2 $f_1 - f_3, f_0 \% (f_7 - f_2), f_4 * f_2, f_5 + f_6$

3 $f_6 / f_9, (f_0 * f_8) - f_3, f_2 + f_5, f_1 \% (f_3 * f_2), f_4 - f_7$

4 $f_2 * f_1, f_3 \% (f_8 - f_5), f_5 / f_4, f_0 + (f_7 - f_3), f_9 - f_6, f_1 * f_3$

5 $f_4 - f_2, f_9 * f_3, f_1 / (f_6 - f_0), f_7 + f_5$

Then Reverse Polish Notations are as follows:

1 $f_0 f_4 *, f_3 f_7 f_1 * +, f_2 f_6 \%, f_1 f_9 /, f_5 f_8 - f_4 *$

2 $f_1 f_3 -, f_0 f_7 f_2 - \%, f_4 f_2 *, f_5 f_6 +$

3 $f_6 f_9 /, f_0 f_8 * f_3 -, f_2 f_5 +, f_1 f_3 f_2 * \%, f_4 f_7 -$

4 $f_2 f_1 *, f_3 f_8 f_5 - \%, f_5 f_4 /, f_0 f_7 f_3 - +, f_9 f_6 -, f_1 f_3 *$

5 $f_4 f_2 -, f_9 f_3 *, f_1 f_6 f_0 - /, f_7 f_5 +$

The final sequences are as follows:

1 $< SOS > \ f_0 \ f_4 \ * \ < SEP > \ f_3 \ f_7 \ f_1 \ * \ + \ < SEP > \ f_2 \ f_6 \ \% \ < SEP > \ f_1 \ f_9 \ / \ < SEP > \ f_5 \ f_8 \ - \ f_4 \ * \ < EOS >$

2 $< SOS > \ f_1 \ f_3 \ - \ < SEP > \ f_0 \ f_7 \ f_2 \ - \ \% \ < SEP > \ f_4 \ f_2 \ * < SEP > \ f_5 \ f_6 \ + \ < EOS >$

3 $< SOS > \ f_6 \ f_9 \ / \ < SEP > \ f_0 \ f_8 \ * \ f_3 \ - \ < SEP > \ f_2 \ f_5 \ + \ < SEP > \ f_1 \ f_3 \ f_2 \ * \ \% \ < SEP > \ f_4 \ f_7 \ - \ < EOS >$

4 $< SOS > \ f_2 \ f_1 \ * < SEP > \ f_3 \ f_8 \ f_5 \ - \ \% \ < SEP > \ f_5 \ f_4 \ / \ < SEP > \ f_0 \ f_7 \ f_3 \ - \ + \ < SEP > \ f_9 \ f_6 \ - \ < SEP > \ f_1 \ f_3 \ * \ < EOS >$

5 $< SOS > \ f_4 \ f_2 \ - \ < SEP > \ f_9 \ f_3 \ * < SEP > \ f_1 \ f_6 \ f_0 \ - \ / \ < SEP > \ f_7 \ f_5 \ + \ < EOS >$

## B  DATASET DETAIL INFORMATION

Table 3: Sensitive features' information

| Dataset | Senstive Task Type | Senstive Feature Name | Senstive Feature Description |
|---|---|---|---|
| German Credit | Classification | famges | User's marital status |
| Housing Boston | Regression | TAX | Property tax rate |
| Uci Credit Card | Classification | EDUCATION | The user's education level |
| Amazon Employee | Classification | ROLE_CODE | Company role code (e.g. Manager) |

We select 8 real datasets for experiments to demonstrate the effectiveness of our method. These public datasets come from UCI (Public, 2022b), Kaggle (Howard, 2022), and OpenML (Public, 2022a), involving classification and regression problems. Four of the datasets are user-related, including German Credit, Housing Boston, Uci Credit Card, and Amazon Employee. We selected some of the information that users may not want to expose as sensitive features, as shown in Table 3. In addition, there are 4 datasets where sensitive features cannot be directly defined, including lymphography, Openml 618, Activity, and AP-omentum-ovary. We selected their first feature as the sensitive feature. Table 4 shows the statistics of the datasets. We randomly split each dataset into two independent sets. The prior 80% is used to build the continuous embedding space and the remaining 20% is employed to test transformation performance. We report the results of five-fold cross-validation.

Table 4: Datasets Statistics. 'C' for classification, and 'R' for regression.

| Dataset | Source | Type-y | Type-z | # Samples | # Features |
|---|---|---|---|---|---|
| Housing Boston | UCIrvine | R | R | 506 | 13 |
| German Credit | UCIrvine | C | C | 1001 | 24 |
| Uci Credit Card | UCIrvine | C | C | 30000 | 25 |
| Amazon Employee | Kaggel | C | C | 32769 | 9 |
| Lymphography | UCIrvine | C | C | 148 | 18 |
| Openml 618 | OpenML | R | R | 1000 | 50 |
| AP Omentum Ovary | OpenML | C | C | 275 | 10936 |
| Activity | UCIrvine | C | R | 10299 | 561 |

# C CONSTRAINED GRADIENT UPDATE

We optimize the latent representation towards better performance and privacy by updating the gradients under progressively tighter constraints. We first use the trained privacy evaluator to evaluate the HSIC of the initial latent variables and sensitive features as the initial privacy constraint $\Psi_{pr}(\hat{E}_p^{min}; s)$. We use a performance estimator to guide the optimization of the latent representation $E_p$ towards better performance. We use privacy constraints to ensure that the updated $\hat{E}_p$ has better privacy under the evaluation of the privacy evaluator. We use projected gradient descent to implement such constrained updates:

$$\hat{E}_p = E_p + \eta proj_{\Psi_{pr}(\hat{E}_p; s) \leq \Psi_{pr}(\hat{E}_p^{min}; s)} \frac{\partial \Psi_{pr}}{\partial E_p} \tag{10}$$

The specific implementation is shown in Algorithm 1.

---

**Algorithm 1** Projected Gradient Descent Optimization with Dynamic Constraints

---

1: Initialize $E_p$ and $E_p^{min}$
2: **for** each iteration $t = 1, 2, \ldots, T$ **do**
3:     Compute gradient $\nabla \Psi_{pr}(E_p)$
4:     Update $E_p \leftarrow E_p + \eta \nabla \Psi_{pr}(E_p)$
5:     **if** $\Psi_{pr}(E_p; s) \leq \Psi_{pr}(E_p^{min}; s)$ **then**
6:         Set $E_p^{min} \leftarrow E_p$
7:     **else**
8:         Initialize $k = 0$
9:         **while** $k < K$ and $\Psi_{pr}(E_p; s) > \Psi_{pr}(E_p^{min}; s)$ **do**
10:             Compute gradient $\nabla \Psi_{pr}(E_p)$
11:             Update $E_p \leftarrow E_p + \eta \nabla \Psi_{pr}(E_p)$
12:             **if** $\Psi_{pr}(E_p; s) \leq \Psi_{pr}(E_p^{min}; s)$ **then**
13:                 Set $E_p^{min} \leftarrow E_p$
14:                 Break
15:             **end if**
16:             $k \leftarrow k + 1$
17:         **end while**
18:     **end if**
19: **end for**

---

In addition, we first select $m$ initial latent representations as seeds for constrained updating to obtain $m$ candidate representations $[\hat{E}_p^1, \ldots, \hat{E}_p^m]$. We adopt the beam search strategy to obtain better representation candidates. Specifically, given an updated embedding $\hat{E}_p$, at step-$t$, we maintain the historical predictions with beam size $b$, denoted as $\{\eta_{<t}^i\}_{i=1}^b$. For the $i$-th beam, the probability distribution of the token identified by the well-trained decoder $\Gamma_d$ at the $t$-th step is $\gamma$, which can be calculated as follows:

$$P_t^i(\gamma) = P_{\Gamma_d}(\gamma | \hat{E}_p, \hat{\eta}_{<t}^i) \cdot P_{\Gamma_d}(\hat{\eta}_{<t}^i | \hat{E}_p), \tag{11}$$

where the probability distribution $P_t^i(\gamma)$ is the continued multiplication of the probability distribution from the previous decoding sequence and the current decoding step. We collect the conditional probability distribution of all tokens for each beam. After that, we append tokens with the top-$b$ probability values to the historical predictions of each beam to form a new set of predictions $\{\hat{\eta}_{<t+1}^i\}_{i=1}^b$. We iteratively conduct this decoding process until reaching the <EOS> token. We then select the transformation sequence with the highest probability value as output. Thus, $T$ enhanced embeddings may produce $T$ transformation sequences $\{\hat{\eta}^i\}_{i=1}^T$. Each sequence is divided into different parts according to the <SEP> token, and we check the validity of each part, removing the invalid ones. Here, the validity checks whether the mathematical compositions represented by the Reverse Polish Notation can be successfully computed to produce a new feature. These valid form the final Reverse Polish Notation sequence $\{\hat{\eta}_p^i\}_{i=1}^T$, which is used to generate the refined feature space $\{\hat{F}^i\}_{i=1}^T$. Finally, we select the feature set with the highest downstream ML performance as the optimal feature space $F^*$.

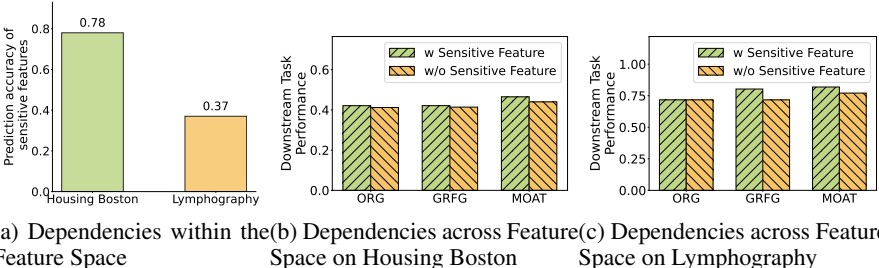

(a) Dependencies within the Feature Space    (b) Dependencies across Feature Space on Housing Boston    (c) Dependencies across Feature Space on Lymphography

Figure 7: Impact of Feature Dependencies

## D EXPERIMENT ENVIRONMENT AND SETUP

All experiments were conducted on the Ubuntu 11.2 operating system, 13th Gen Intel(R) Core(TM) i9-13900KF CPU, and 2 NVIDIA GeForce RTX 4090 GPUs, with the framework of Python 3.10.14 and PyTorch 2.3.1. We ran the privacy-awared feature space construction for 10 epochs. We randomly shuffled each Reverse Polish Notation 12 times to increase data diversity and volume. We adopted a single-layer LSTM as the encoder and decoder backbones and utilized 3-layer feed-forward networks to implement the predictor. The hidden state sizes of the encoder, decoder and predictor are 64, 64, and 200, respectively. The embedding size of each feature ID token and operation token was set to 32. The autoencoder and estimator are trained for 2000 epochs. When implementing the constraint updates, we allow the projected gradient descent step to be executed at most 100 times. To facilitate the adjustment of weights, we used the following form when implementing equation (7):

$$\mathcal{R}(F_i, y, s) = (1 - \alpha)\mathbb{I}(F_i; y) - \alpha H\hat{S}IC(F_i; s). \tag{12}$$

The value range of $\alpha$ is (0,1). We train our model with $\alpha = [0.1, 0.3, 0.5, 0.7, 0.9]$ and select the best result to report. For the methods in the baseline, we train them according to the parameters given by the authors in the original paper. When reporting accuracy on downstream tasks and sensitive features, we use 5-fold cross-validation.

## E SUPPLEMENTARY EXPERIMENTS

### E.1 IMPACT OF FEATURE DEPENDENCIES

When considering privacy from a data perspective, simply deleting sensitive features can cause uncontrollable negative impacts due to the complex feature dependencies within and across feature spaces. Dependencies within the feature space mean that it's not enough to simply address the sensitive features themselves. Other features may inadvertently expose sensitive information, requiring a more comprehensive approach. Dependencies across feature space refer that sensitive features often contribute to the construction of new features, creating associations within the transformed feature space. Directly removing sensitive features may cause multiple valuable new features to become unavailable, significantly reducing downstream tasks' performance. The experimental results in Figure 7 illustrate the necessity of utilizing sensitive features for feature transformation from the perspective of the overall feature space . Figure 7(a) shows the use of other features to predict sensitive features in the original dataset. The results show that sensitive information is at risk of indirect exposure. In the Housing Boston dataset, the accuracy of using other features to predict sensitive features is as high as 78%. This shows that when considering privacy from a data perspective, it is not enough to only process sensitive features themselves. Figure 7(b) and Figure 7(c) Demonstrates the impact of feature dependencies across space. We delete sensitive features in the original data set and use two SOTA feature transformation methods for feature transformation. We report the final downstream task accuracy when sensitive features are involved and the downstream task accuracy when no sensitive features are involved. The results indicate that feature transformation may further exacerbate the negative degradation caused by removing sensitive features. For example, in the Lymphography dataset, deleting sensitive features in the original data will not cause performance

Table 5: Estimator Performance

| Dataset | Performance Evaluator | Privacy Evaluator |
|---------|----------------------|-------------------|
| Housing Boston | 0.6898 | 0.8510 |
| German Credit | 0.6565 | 0.8281 |
| Uci Credit Card | 0.6313 | 0.7219 |
| Amazon Employee | 0.7085 | 0.6082 |
| Lymphography | 0.6619 | 0.6031 |
| Openml 618 | 0.6578 | 0.6611 |
| AP Omentum Ovary | 0.5942 | 0.7153 |
| Activity | 0.5479 | 0.5515 |

Table 6: Robustness check with distinct ML models

| DownStream Method | Task | Lymphography | | | | Housing Boston | | | |
|---|---|---|---|---|---|---|---|---|---|
| | | ORI | GRFG | MOAT | Ours | ORI | GRFG | MOAT | Ours |
| RandomForest | Downstream Task | 0.7175 | 0.8133 | 0.8185 | 0.7895 | 0.4012 | 0.4212 | 0.4648 | 0.4574 |
| | Sensitive Feature | 0.4445 | 0.6323 | 0.5100 | 0.7294 | 0.1630 | 0.1109 | 0.0391 | 0.1747 |
| | Average | 0.5810 | 0.7228 | 0.6642 | **0.7595** | 0.2821 | 0.2661 | 0.2520 | **0.3161** |
| MLP | Downstream Task | 0.7518 | 0.8265 | 0.7552 | 0.7842 | 0.4522 | 0.5153 | 0.5131 | 0.5016 |
| | Sensitive Feature | 0.6756 | 0.3841 | 0.4256 | 0.6996 | 0.6756 | 0.6050 | 0.5000 | 0.7050 |
| | Average | 0.7137 | 0.6053 | 0.5904 | **0.7419** | 0.5639 | 0.5602 | 0.5066 | **0.6033** |
| SVM | Downstream Task | 0.7461 | 0.8114 | 0.7518 | 0.7861 | 0.0226 | 0.3881 | 0.0296 | 0.0896 |
| | Sensitive Feature | 0.6330 | 0.4970 | 0.6375 | 0.6623 | 0.9147 | 0.6825 | 0.8965 | 0.9136 |
| | Average | 0.6896 | 0.6542 | 0.6947 | **0.7242** | 0.4686 | 0.5353 | 0.4631 | **0.5016** |
| GradientBoosting | Downstream Task | 0.7861 | 0.8488 | 0.7895 | 0.8285 | 0.4353 | 0.6042 | 0.4594 | 0.4552 |
| | Sensitive Feature | 0.5715 | 0.0035 | 0.0723 | 0.5874 | 0.2100 | 0.0032 | 0.1724 | 0.1946 |
| | Average | 0.6788 | 0.4262 | 0.4309 | **0.7080** | 0.3227 | 0.3037 | 0.3159 | **0.3249** |

degradation in downstream tasks, but if the sensitive features do not participate in feature transformation, the performance degradation will be significant. This illustrates that simple deletion or preprocessing of sensitive features may lead to suboptimal performance of downstream tasks. It is necessary to involve it in the feature conversion process.

### E.2 PERFORMANCE OF EVALUATORS

In our approach, there are two evaluators that use latent variables to evaluate downstream task performance and sensitive information exposure risk, respectively. In this section, we show the performance of these two evaluators during training. Considering that we only need to update the gradient in the correct direction based on the relative size and tighten the privacy constraint, we use pairwise accuracy as the measurement metric. The formula for pairwise accuracy is given by:

$$\text{Pairwise Accuracy} = \frac{1}{\binom{n}{2}} \sum_{i=1}^{n-1} \sum_{j=i+1}^{n} 1\left((y_i > y_j) \wedge (\hat{y}_i > \hat{y}_j) \vee (y_i < y_j) \wedge (\hat{y}_i < \hat{y}_j)\right) \quad (13)$$

where 1 is the indicator function, and $\binom{n}{2}$ represents the number of sample pairs. As shown in Table 5, in our experiments, both evaluator tasks achieve usable accuracy.

### E.3 ROBUSTNESS CHECK WITH DISTINCT ML MODELS

We replaced the downstream ML models with RandomForest, MLP (Rumelhart et al., 1986), Support Vector Machine (SVM) (Cortes, 1995), GradientBoosting (Friedman, 2001) to observe the variance of model performance, respectively. Table 6 shows the comparison results on Lymphography

Table 7: Time Cost

| Dataset | # Samples | # Features | Privacy-Awared Constructio(s) | Model Training (s/epoch) | Latent Rep Updated w/o PGD (s) | Latent Rep Updated (s) |
|---|---|---|---|---|---|---|
| German Credit | 1001 | 24 | 123 | 1.9 | 11.1 | 15.2 |
| Housing Boston | 506 | 13 | 101 | 2 | 9.7 | 12.6 |
| Uci Credit Card | 30000 | 25 | 39908 | 1.9 | 10.4 | 12 |
| Amazon Employee | 32769 | 9 | 50569 | 2 | 9.9 | 11.2 |
| Lymphography | 148 | 18 | 59 | 1.7 | 9.1 | 10.9 |
| Openml 618 | 1000 | 50 | 2155 | 2.7 | 15.7 | 17.6 |
| Activity | 10299 | 561 | 58519 | 3.8 | 20.4 | 22.7 |
| AP Omentum Ovary | 275 | 10936 | 2458 | 2.3 | 15.2 | 16.7 |

and Housing Boston. We report our model and the two strongest baseline performances. We use the same evaluation metrics as in Table 1. The results show that PFT performs optimally in terms of both performance and privacy average metric when the downstream task models are different.

### E.4 TIME COMPLEXITY

PFT achieves the dual goals of performance and privacy without introducing complex model structures and additional storage space, so its space complexity is similar to that of current feature transformation models. In this section, we mainly show the time cost of our model. We show the time cost of each link of our method on different datasets in Table 7. Privacy-Awared Constructio represents the total time cost of our privacy-aware feature space construction. Model Training represents the time required for each epoch in the training of the autoencoder. Latent Rep Updated w/o PGD represents that we remove the asymptotic constraints and only optimize towards better performance in the regeneration phase. Latent Rep Updated represents the constrained latent representation update used in our method.

As shown in the table, different datasets have large time differences in the privacy-aware feature space construction. This time difference is mainly determined by the number of samples. Through our further observation, the time overhead at this time is mainly caused by evaluating the new feature space on the downstream task model. There is room for optimization at the implementation level. In addition, during the model training process, the time cost on different datasets is similar. This is because we represent the feature space in the latent space, and the representation length in the latent space is fixed, which prevents the model training from experiencing a catastrophic increase in training cost due to the increase in features. In addition, the introduction of constraints requires additional time to perform projected gradient descent, but the additional time cost is 2s on average, which is within an acceptable range.

