# OpenReview forum: "Privacy Preserving Generative Feature Transformation"
_ICLR.cc/2025/Conference — ICLR 2025 Conference Withdrawn Submission_

### Official Review · Reviewer_c5qP · 2024-10-20

**Soundness:** 2
**Presentation:** 2
**Contribution:** 2
**Rating:** 3
**Confidence:** 4

**Summary:**

The paper presents a two-phase Privacy-preserving Feature Transformation (PFT) framework that integrates concepts from generative AI, the privacy information bottleneck approach, and gradient-based optimization. In the first phase, multi-agent reinforcement learning is utilized to generate privacy-aware feature sets, where the agents aim to maximize the mutual information between the transformed features and downstream tasks while minimizing the mutual information between the transformed features and sensitive attributes. In the second phase, generative models are employed to create privacy-preserving feature spaces that also maintain utility. This is achieved through a constrained gradient update, where the objective function optimizes performance metrics while the constraints ensure that the sensitive attributes become less exposed with each update iteration. Experiments on real-world datasets show that PFT enhances the balance between performance and privacy metrics, consistently achieving better results across all datasets based on the average of privacy and utility performance metrics.

**Strengths:**

1) This work addresses the challenge of privacy-preserving feature transformation, demonstrating that the two-phase PFT framework outperforms non-private feature transformation methods and even simple classic differentially private adaptations of these methods.

2) The paper introduces a novel combination of generative AI, the privacy information bottleneck approach, and gradient-based optimization to develop the framework.

3) The first three figures (Figures 1-3) effectively illustrate the concept of the paper. While some parts of the main text lack clarity, these figures assist in understanding the framework.

4) The diverse experiments, particularly those in Section 4.3.1 (correlation of features with downstream tasks and sensitive features) and Section 4.4 (ablation study), are well-motivated and provide valuable insights into the effectiveness of the PFT framework.

**Weaknesses:**

1) On page 4, it is stated that PFT maximizes the incremental performance of the feature space, which serves as a lower bound of the mutual information. However, the paper does not provide a justification for why it optimizes this lower bound instead of the mutual information itself. Additionally, in Equation (3), the derivation of this lower bound has issues. Specifically, inequality (c) is unclear, as the definition of information entropy suggests that for inequality (c) to hold, the following must be true:
\begin{equation*}
   \sum\limits_{F_i, y} p(F_i, y) \log(p(F_i \mid y)) \geq \sum\limits_{F_i} p(F_i) \log(p(F_i \mid y))
\end{equation*}
   However, this transformation generally does not hold, as the left-hand side uses the joint probability while the right-hand side uses the marginal probability.

2) It is mentioned in the appendix, that the parameter $\alpha$ that controls the trade-off between privacy and utility in the experiments is chosen from different values, with the best performance reported for PFT. However, for the baseline methods, parameters are taken directly from the original papers. This approach may not provide a fair comparison, as PFT's parameters are fine-tuned to maximize the average objective function reported in Tables 1 and 2, whereas the baseline methods, particularly the differentially private benchmarks, are not similarly optimized for this objective.

3) There are some misrepresentations and errors, particularly regarding the minimization and maximization of objectives. In Equation (1), while $\mathcal{L}$ appears to represent the loss function, it seems that the $ \min $ and $\max $ operators may have been interchanged. The utility loss function should be minimized, while the loss function for predicting the sensitive attribute should be maximized.

**Questions:**

1) While it is mentioned in the paper that privacy might be unmeasurable for the specific task of privacy-preserving feature transformation, is there a reason why differential privacy is not considered as a standard for measuring privacy?

2) I cannot follow Equation (3)(c). Could you please provide a justification for it? Additionally, I suggest using an index in Equation (3) for the summation, as it is unclear what the summation is over (which parameter).

3) In line 283, what exactly is $ v $? I know that in the appendix it is mentioned that pairwise accuracy can be used as a performance metric, but it is not clear from the main text how $\Psi_{pe} $ is trained. Could you please explain this further?

Here is a minor comment:

4)  In line 128: "We minimize the mutual information between the new feature space and downstream tasks while minimizing the mutual information between the new feature space and sensitive features." I believe "minimization" should be changed to "maximization."

If my concerns are addressed, I would be happy to consider increasing my score.

---

> ### Author Response · Authors · 2024-12-02
>
> Thank you for acknowledging the importance of our motivation, the validity of our approach, the clarity of our figures, and the solid techniques and diverse experiments presented in our work.
>
> ---
>
> ## W1.1: Why Maximizing the Mutual Information Lower Bound Rather Than Itself
>
> Maximizing the lower bound of mutual information is a standard approach in machine learning and information theory [1-3]. Directly computing mutual information (MI) is often computationally intractable or statistically challenging, particularly in high-dimensional or continuous settings. By leveraging a lower bound, we achieve a tractable and effective optimization strategy while maintaining theoretical rigor.
>
> ---
>
> ## W1.2 & Q2: Explaining Step (c) of Equation 3
>
> Thank you for pointing out the typo. Our textual description is correct, but the equation had a typo. We have revised the formula as follows:
>
> $$
> \begin{aligned}
>     \mathbb{I}(F_{i}; y) & \overset{(a)}{=} H(F_{i}) - H(F_{i}|y) \overset{(b)}{\geq} -H(F_{i}|y) \overset{(c)}{=} \sum p(F_{i}, y) \log{(p(F_{i}|y))} \\
>     & \overset{(d)}{\geq} \log{(p(F_{i}|y))} \overset{(e)}{=} \log{(\phi(\mathcal{D}(F_{i})))}
>     \overset{(f)}{\geq} \log{(\phi(\mathcal{D}(F_{i})))}
>     - \log{(\phi(\mathcal{D}(F_{i-1})))},
> \end{aligned}
> $$
>
> ---
>
> ## W2: Explaining Why Experimental Comparisons Are Fair
>
> Our optimization is based on constrained optimization. When comparing our method with baseline methods, we optimize both our methods and baselines to achieve the best predictive accuracy for a downstream task (e.g., heart attack risk prediction). Then, we report the privacy protection effectiveness of our method and baselines under the maximum downstream task predictive accuracy. Therefore, our comparisons are fair.
>
> ---
>
> ## Q1: Why Not Using Differential Privacy as an Evaluation Criterion
>
> Differential privacy (DP) typically requires introducing randomness (e.g., noise injection, perturbation) into the transformation process to ensure that even minor changes to the input do not significantly alter the output distribution.
>
> Our and other baseline feature transformation processes do not rely on noise injection, making them incompatible with direct evaluation via a DP budget $\( \epsilon \)$.
>
> Instead, we use **Attribute Inference Attacks** [4-6] to empirically assess privacy leakage. Specifically:
> - Non-sensitive features from the transformed dataset are used to predict sensitive features.
> - The model’s prediction performance on the test set serves as a measure of privacy leakage.
>
> ---
>
> ## Q3: What Exactly is $v$? How $\Psi_{pe}$ is Trained
>
> The variable $v$ represents the accuracy of a downstream model on a test set, used to evaluate the performance of a given feature space.
>
> During knowledge collection, $v$ is recorded for each feature space. In training, $\Psi_{pe}$, a simple MLP, predicts the downstream performance $\hat{v}$ using the latent space representation.
>
> The loss for optimizing $\Psi_{pe}$ is the MSE between the true performance $v$ and the predicted performance $\hat{v}$.
>
> ---
>
> ## Q4: In Line 128: "Minimization" Should Be Changed to "Maximization"
>
> Thank you for pointing out the typo. We have corrected the text to change "minimization" to "maximization" as intended.
>
> ---
>
> ## References
>
> [1] Barber, David, and Felix Agakov. "The im algorithm: a variational approach to information maximization." Advances in neural information processing systems 16.320 (2004): 201.
> [2] Oord, Aaron van den, Yazhe Li, and Oriol Vinyals. "Representation learning with contrastive predictive coding." arXiv preprint arXiv:1807.03748 (2018).
> [3] Tschannen, Michael, et al. "On mutual information maximization for representation learning." arXiv preprint arXiv:1907.13625 (2019).
> [4] Fredrikson, Matt, Somesh Jha, and Thomas Ristenpart. "Model inversion attacks that exploit confidence information and basic countermeasures." Proceedings of the 22nd ACM SIGSAC conference on computer and communications security. 2015.
> [5] Yeom, Samuel, et al. "Privacy risk in machine learning: Analyzing the connection to overfitting." 2018 IEEE 31st computer security foundations symposium (CSF). IEEE, 2018.
> [6] Zhao, Benjamin Zi Hao, et al. "On the (in) feasibility of attribute inference attacks on machine learning models." 2021 IEEE European Symposium on Security and Privacy (EuroS&P). IEEE, 2021.

---

### Official Review · Reviewer_Ek3q · 2024-11-01

**Soundness:** 2
**Presentation:** 2
**Contribution:** 2
**Rating:** 3
**Confidence:** 4

**Summary:**

This paper proposes a privacy-preserving feature transformation pipeline. The pipeline consists of two parts:
1. privacy-aware knowledge acquisition: A reinforcement learning framework is used to transform the feature set in a way that maximizes the mutual information (MI) between the transformed features and the downstream task and minimizes the MI between the transformed features and the sensitive features.
2. privacy-preserving feature space generation: The transformed feature set from part 1 is treated as a sequence of tokens and map it to a latent representation using a sequential encoder. A decoder is used to map this representation back to the feature space. Performance and privacy evaluators are built to estimate the utility and privacy risks from the latent representation. The latent representation is modified using a constrained update rule in a way that improves utility while reducing the privacy leakage.

Evaluations are performed on multiple datasets to show that the proposed method can provide high performance on downstream tasks while making it harder to recover sensitive features.

**Strengths:**

1. The paper studies an important problem of privacy-preserving feature transformation.
2. Experiments seem extensive.

**Weaknesses:**

1. The proposed method does not provide strong theoretical privacy guarantees. Note that empirical evaluations of privacy is insufficient to claim that the proposed feature transformation technique can protects against all future privacy leakage attacks.
2. In table 1, the proposed method seems to have high "SF" (prediction accuracy on sensitive features). Doesn't this mean that the proposed method is worse-off in terms of privacy compared to other methods? I also don't understand why a higher "Avg"  of "DT" and "SF" is desirable.

**Questions:**

1. Is there a mistake in Eq. 9? $\psi_{pe}$ does not appear in either the objective or the constraint.
2. Can you provide any guarantees for worst case privacy leakage for your proposal?
3. It's not clear why the proposed method has better privacy from the evaluations (see weakness #2)

---

> ### Author Response · Authors · 2024-12-02
>
> Thank you for acknowledging the importance of our problem and the extensiveness of our experiments.
>
> ## W1 & Q1 & Q2: How to Ensure Privacy and Address Worst-Case Privacy Leakage
>
> - **Measurement**:
>   We employed the **information bottleneck**, a theoretically proven principle, to ensure that in a dataset, label information is preserved while private information is filtered out.
>
> - **Objective Functions and Constrained Optimization**:
>   Our objective function maximizes downstream task predictive accuracy while minimizing privacy exposure risk. We solve this as a constrained optimization problem, providing a clear and tangible approach to identify a feasible optimal solution.
>
> - **Experiments**:
>   We developed comprehensive experiments to examine various aspects (performance, privacy protection, cost, rationality, etc.) of our method. Our framework demonstrates strong empirical performance in mitigating privacy risks across diverse datasets.
>
> ---
>
> ## W2: Definition of SF in Evaluating Indicators
>
> We have defined **“SF”** in Section 4.1.2. SF is a loss measure for inferring sensitive features:
> - For classification tasks, **SF** is $\(1 - \text{F1 score}\)$.
> - For regression tasks, **SF** is Relative Absolute Error (RAE).
>
> Thus, when SF is large, the downstream model fails to infer sensitive attributes, meaning less privacy is exposed.
>
> **DT** is an accuracy measure for predicting downstream tasks. The larger the **DT** value, the better the performance of our solution.
>
> To evaluate both the performance of inferring sensitive features and predicting downstream tasks, we use the **Avg** metric, which considers the trade-offs between utility (**DT**) and privacy protection (**SF**).
>
> ---
>
> ## Q1: Doubts About the Role of $\Psi_{pr}$
>
> Thanks for pointing out the typo. We have fixed Eq(9) as follows:
>
> $$
> \begin{aligned}
>     & \hat{E_p} = E_p + \eta \frac{\partial \Psi_{pe}}{\partial E_p} \\
>     & \text{s.t.} \quad \Psi_{pr}(\hat{E_p}; s) \leq \Psi_{pr}(\hat{E_p}^{\text{min}}; s),
> \end{aligned}
> $$
>
> This correction ensures that the representation aligns with the intended objective.

---

### Official Review · Reviewer_BNr5 · 2024-11-01

**Soundness:** 3
**Presentation:** 2
**Contribution:** 3
**Rating:** 5
**Confidence:** 2

**Summary:**

This paper proposes a framework for performing feature transformation with privacy preservation. The framework first extracts privacy-aware knowledge from the dataset using information bottleneck-guided multi-agent reinforcement learning. It then generates a privacy-preserving feature space. The resulting feature space not only maintains privacy but also enhances the accuracy of downstream tasks.

**Strengths:**

1. The paper presents a valuable idea for feature transformation that effectively incorporates privacy preservation.
2. It includes clear visual explanations and sufficient mathematical details. The authors also provide comprehensive experimental results that demonstrate the effectiveness of their approach.

**Weaknesses:**

1. The paper lacks sufficient background information on reinforcement learning and the DQN structure, which could make it difficult for readers unfamiliar with these concepts to fully understand the framework.

2. The presentation could be improved. The related work section would be more effective if placed earlier in the paper, and some figures, such as Figure 5, are missing important details. Additional explanations are needed to make the visual information clearer and more informative.

**Questions:**

1. In Section 3.2.1, what specific DQN structure is used? It would be helpful to have more background information on this. Could you also explain why you chose this particular structure for your framework?

2. What does the X-axis represent in Figure 5? Could you provide more details on how this figure was generated and explain what it illustrates?

3. I assume that "sensitive features" refer to features that contain private information. How accurately does your method identify these sensitive features? In the evaluation section, could you explain in more detail how you assess the accuracy of sensitive feature detection?

---

> ### Author Response · Authors · 2024-12-02
>
> Thank you for acknowledging that the idea is valuable, the explanations are clear, the details are sufficient, and the experiments are comprehensive.
>
> ## W1: Adding Background of Reinforcement Learning  & Q1: DQN Structure
>
> The task of Section 3.2.1 is to collect training data of transformed feature sets (e.g., transformed patient table data), corresponding predictive performance (e.g., RMSE of heart attack risk regression), and corresponding privacy exposure risk (e.g., HSIC between gender/race and non-sensitive features).
>
> Training data collection can be manually collected by students/volunteers. However, since the possibilities of combinations of feature transformation are large, instead of manual collection, we use reinforcement learning agents to explore various transformed feature sets and test corresponding predictive performance and privacy exposure risk for us. This is why we call RL a training data collector.
>
> DQN is one of the most famous value learning-based RL methods for discrete actions. We use the widely used classic DQN structure by using a simple vanilla neural network as the state-action value estimation function. The role of DQN is to explore and collect training data to learn privacy-aware feature transformation.
>
> As stated in the paper, we employ a standard DQN structure without additional modifications. This simplicity ensures the interpretability and reproducibility of our approach.
>
> ---
>
> ## W2: Place the Related Work Section Earlier in the Paper
>
> In our offline revised version, we have moved RELATED WORK to Section 2.
>
> ---
>
> ## Q2: The X-Axis Representation in Figure 5
>
> Figure 5 shows the correlation between HSIC scores and Sensitivity (how accurately we can use a model to infer sensitive features with a model on a transformed feature set) over a series of transformed feature sets explored by reinforcement learning agents.
>
> Thus, the X-axis represents the index of a series of transformed feature sets explored by RL, for instance:
> - Transformed feature set #1
> - Transformed feature set #2
> - Transformed feature set #3
> ...
>
> ---
>
> ## Q3: Identification and Evaluation of Sensitive Features
>
> 1. In our experiments, for instance, sensitive features are gender and race.
>
> 2. We didn’t identify which features are sensitive. Since we know the names (e.g., gender and race) of features, we directly predefine gender and race as sensitive features.
>
> 3. Since we predefined which features are sensitive, we didn’t need to detect sensitive features.
>
> 4. Our focused task is to evaluate how our transformed feature set can improve prediction accuracy while avoiding sensitive features from being inferred, instead of identifying or detecting sensitive features.

---

### Official Review · Reviewer_T7Lm · 2024-11-06

**Soundness:** 3
**Presentation:** 2
**Contribution:** 2
**Rating:** 3
**Confidence:** 2

**Summary:**

This paper studies techniques to transform the feature space of the dataset while preserving privacy. A new method is proposed that consists of two steps 1) privacy-aware knowledge acquisition using multi-agent reinforcement learning  2) privacy-preserving feature space generation by mapping from feature space to a privacy-preserving latent space. The main objective function is to optimize of the transformation space to maximize the mutual information between new feature and downstream task while minimizing the mutual information between the private task and the new feature space.

**Strengths:**

The problem of optimizing over the transformation space is interesting and the objective function seems reasonable.

**Weaknesses:**

The presentation and the problem formulation are not clear to me.
For example, the motivation of the problem is deferred to Appendix E, while I believe this is important to explain why targeting this problem.
Also, I read Appendix E, and couldn't clearly understand the challenging problem. It seems in Figure 7 that downstream task performance w/o sensitive features is close to the now w sensitive features.

The problem statement is not clear as well. For example, what is the original feature F? is it real space of d dimensions? also what are A_pr and A_pe models? how they are different? Also could you please correlate this problem statement to a real scenario?

**Questions:**

see above

---

> ### Author Response · Authors · 2024-12-02
>
> Thank you for acknowledging the interesting formulation and the reasonable objective function design.
>
> ## W1.1: Clarity of Problem Formulation
>
> Our AI task is privacy-aware feature transformation. Our task is to transform a given feature set into a new feature set, such that:
> 1. we can use the transformed feature set to improve predictive accuracy;
> 2. we cannot use the transformed feature set to infer sensitive features.
>
> ---
>
> ## W1.2: Experiments in Appendix E Should Be Moved to the Main Text
>
> Thanks for highlighting the importance of Appendix E. Due to space limits, the experimental motivation example is moved to Appendix E.
>
> ---
>
> ## W2: Figure 7 that downstream task performance w/o sensitive features is close to the now w sensitive features
>
> This experiment (Line 916-950)  is to demonstrate the strengths of privacy-aware feature transformation formulation:
> 1. our formulation (i.e., using transformed data) can not just leverage sensitive features to improve predictive accuracy, but also protect sensitive features from being inferred.
> 2. Whereas, there are two weaknesses of direct sensitive feature deletion:
>    1) although deleted, sensitive features can be inferred by non-sensitive features;
>    2) direct sensitive features deletion can degrade the predictive power of a feature set.
>
> ---
>
> ## W3: Problem Statement Clarification and Real Application Scenario
>
> With a data mining academic background, our problem statement is clear for the following reasons:
> 1. Feature transformation is to transform a given feature set to a new feature set.
> 2. The original feature set refers to the given, old features of a table before transformation.
> 3. The new feature set refers to the transformed, new features of the table after transformation.
> 4. $A_{pe}$ is the model that predicts a downstream task (e.g., yes-no binary classification of heart disease).
> 5. $A_{pr}$ is the model that infers sensitive features using non-sensitive features.
>
> Considering heart attack risk regression, given the tabular data of patients with the sensitive features such as race and gender, the privacy-aware feature transformation is to transform the feature set of this tabular data into a new feature set, so that:
> 1) we maximize regression accuracy of heart attack risks;
> 2) we minimize the risk of using transformed features to infer race and gender.

---

### Note · Authors · 2024-12-02

I have read and agree with the venue's withdrawal policy on behalf of myself and my co-authors.